# Going Beyond 1-WL Expressive Power With 1-Layer Graph Neural Networks

## Abstract

Graph neural networks have become the *de facto* standard for representational learning in graphs, and have achieved SOTA in many graph-related tasks such as node classification, graph classification and link prediction. However, it has been shown that the expressive power is equivalent maximally to Weisfeiler-Lehman Test. Recently, there is a line of work aiming to enhance the expressive power of graph neural networks. In this work, we propose a more generalized variant of neural Weisfeiler-Lehman test to enhance structural representation for each node in a graph to uplift the expressive power of any graph neural network. It is shown theoretically our method is strictly more powerful than 1&2-WL test. The Numerical experiments also show that our proposed method outperforms the standard GNNs on almost all the benchmark datasets by a large margin in most cases with significantly lower running time and memory consumption compared with other more powerful GNNs.

## 1 Introduction

Graph-structured data is ubiquitous in many real-world applications ranging from social network analysis Fan et al. (2019), drug discovery Jiang et al. (2020), personalized recommendation He et al. (2020) and bioinformatics Gasteiger et al. (2021). In recent years, Graph Neural Networks (GNNs) have seized increasing attention due to their powerful expressiveness and have become dominant approaches for graph-related tasks. Message Passing Graph Neural Networks (MPGNNs) are the most common types of GNNs due to their efficiency and expressivity. MPGNNs can be viewed as a neural version of the 1-Weisfeiler-Lehman (1-WL) algorithm Weisfeiler & Leman (1968), where colors are replaced by continuous feature vectors and neural networks are used to aggregate over node neighborhoods Morris et al. (2019). By iteratively aggregating neighboring node features to the center node, MPGNNs learn node representations that encode their local structures and feature information. A graph readout function can be further leveraged to pool a whole-graph representation for downstream tasks such as graph classification.

Despite the success of MPGNNs, it is proved in some recent literatures that the expressive power of MPGNNs is bounded by 1-WL isomorphism test (Morris et al., 2019; Xu et al., 2018a), i.e, standard MPGNNs or 1-WL GNNs cannot distinguish any (sub-)graph structure that 1-WL cannot distinguish such as for any two n-node r-regular graphs, standard MPGNNs will output the same node representation.

Since then, a few works have been proposed to enhance the expressivity of MPGNNs. Methods proposed by (Morris et al., 2019; Chen et al., 2019; Maron et al., 2019) aim at approximating high-dimensional WL tests. However, these methods require learning all node tuples, which are computationally expensive and not able to scale well to large-scale graphs. Another line of works augment node features to enhance the expressive power of GNNs. E.g., works proposed by (Loukas, 2019; Sato et al., 2021) inject one-hot features or random features to each node of a graph, while other works incorporate structural features to enhance expressivity of GNNs such as distance-based features (Zhang & Chen, 2019; Li et al., 2020) and counting features of certain substructures Bouritsas et al. (2022). More recently, (Zhang & Li, 2021; Zhao et al., 2021) propose to leverage subgraph information that cannot be captured by 1-WL test to infer node representations. Concretely, instead of hashing the direct neighborhood information in 1-WL test, these methods hash the subgraph information, and therefore inject additional structural information in the learning process. These

methods are able to strike a balance between effectiveness and running time complexity. However, scalability and memory consumption are still an issue as these methods need to materialize all the subgraphs into GPU memory.

Accordingly, in this paper we tackle the above defects by proposing a lightweight module which is an extension of neural Weisfeiler-Lehman test to extract meaningful structural representations, which can be leveraged alone or plug into any MPGNN to enhance its expressive power. Our proposed method generalizes a rooted subtree by encoding a multi-hop multi-color rooted subtree, which induces a different message passing function. It is shown theoretically and empirically that our method is strictly more powerful than 1&2-WL test with significant reduction in computational complexity and memory consumption with comparable predictive performance or even superior to previous methods.

Our main contributions are summarized as follows:

**(1) New methodologies.** We develop a more generalized variant of neural WL test, where the message passing function induces a multi-hop multi-color rooted subtree instead of a rooted subtree. Our proposed methods enjoys high flexibility and can be leveraged alone or equipped with any graph neural network.

**(2) Theoretical justification.** We show our method is provably more expressive than 1-WL GNNs with only 1 iteration of message passing.

**(3) High efficiency.** Our method can be equipped with any base graph neural network, incurring almost no additional memory consumption while boosting the performance of the base GNN significantly.

**(4) Superior performance.** We conduct extensive experiments in a wide variety of datasets with different tasks. Empirically our approach outperforms all the baseline GNNs by a large margin in most cases.

## 2 PRELIMINARY

We begin by introducing our notations, followed by presenting the concept of WL test and message passing graph neural network framework.

### 2.1 NOTATION

A graph can be represented as $\mathcal{G} = (\mathcal{V}, \mathcal{E})$, where $\mathcal{V} = \{v_1, \ldots, v_n\}$ is the node set and $\mathcal{E} \subseteq \mathcal{V} \times \mathcal{V}$ is the edge set. $\mathcal{X} = \{x_v \mid \forall v \in \mathcal{V}\}$ is the node feature matrix and $\mathcal{F} = \{e_{uv} \mid \forall e_{uv} \in \mathcal{E}\}$ denote the edge feature matrix. The $k$-hop neighborhood of a node $v \subseteq \mathcal{V}$ is the set of nodes whose distance (shortest path) to $v$ is no greater than $k$ and is denoted as $\mathcal{N}_{\leq k}(v)$, furthermore we denote $\mathcal{N}_k(v)$ to be the $k$-th hop neighbors of node $v$. Given a set of nodes $\tilde{\mathcal{S}} \subseteq \mathcal{V}$, the subgraph induced by $\mathcal{S}$ is a graph that has nodes in $\mathcal{S}$ and every endpoint of the edges is in $\mathcal{S}$. The $k$-hop neighborhoods of node $v$ constitute an induced subgraph denoted by $\mathcal{G}_v^k$. We further denote $\mathcal{D}$ and $\mathcal{A}$ to be the diagonal degree matrix and adjacency matrix of $\mathcal{G}$ respectively, and $\hat{A}_k$ to be the $k$-hop neighborhood matrix of graph $\mathcal{G}$. $\hat{A}_k(i)$ outputs the non-zero entries of the $i$-th node whose distance to it equals to $k$.

### 2.2 WEISFEILER-LEHMAN TEST

WL test is a family of very successful algorithmic heuristics used in graph isomorphism problems. 1-WL test, being the simplest one in the family, works as follows - each node is assigned the same color initially, and gets refined in each iteration by aggregating information from their neighbors' states. The refinement stabilizes after a few iterations and the algorithm outputs a representation of the graph. Two graphs with different representations are not isomorphic. The test can uniquely identify a large set of graphs up to isomorphism (Babai & Kucera, 1979), but there are simple examples where the test tragically fails—for instance, two regular graphs with the same number of nodes and same degrees cannot be distinguished by the test. As a result, a natural extension to 1-WL test is $k$-WL test which provides a hierarchical testing process by keeping the state of $k$-tuples of nodes.

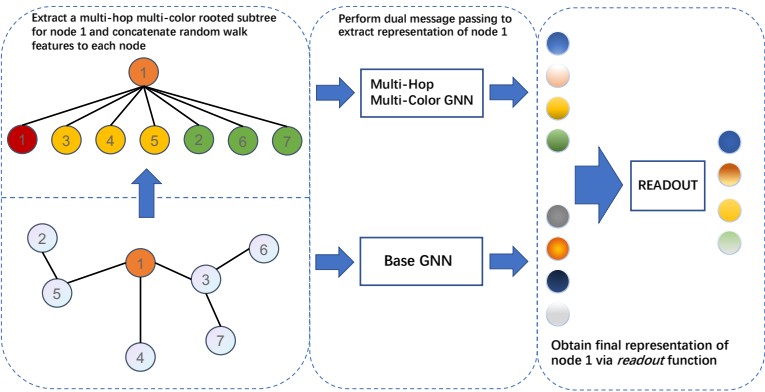

Figure 1: An illustration of the working procedure for GNN-M$^2$HC-RW where the root node is node 1. GNN-M$^2$HC-RW first concatenate the random walk features to every node in the graph, then build a multi-hop multi-color rooted subtree (in this case a 2-hop 2-color rooted subtree), followed by performing message passing induced by standard rooted subtree and multi-hop multi-color rooted subtree. A readout function is then leveraged to summarize the node representations from the dual message passing procedure to get the final representation of node 1.

### 2.3 GRAPH NEURAL NETWORKS

In this paper, we focus on message passing GNNs (MPGNNs). For a MPGNN, the goal is to learn meaningful node representation $h_v$ based on iterative aggregation of local network neighborhoods. The $t$-th iteration of message passing can be written as:

$$a_v^{(t)} = \text{AGG}^{(t)} \left( \left\{ h_u^{(k-1)}, e_{uv}^{t-1} | u \in \mathcal{N}(v) \right\} \right), \quad h_v^{(t)} = \text{UPDATE}^{(t)} \left( h_v^{(t-1)}, a_v^{(t)} \right), \quad (1)$$

where $h_u^{t-1}$ denotes the node $u$'s representation at time stamp $t-1$, $e_{uv}^{k-1}$ is the edge feature of $e_{uv} \subseteq \mathcal{E}$, and $\mathcal{N}(v)$ returns the set of neighbors of node $v$. $\text{AGG}^{(t)}$ and $\text{UPDATE}^{(t)}$ are the aggregation and update functions at time stamp $t$, respectively. After $T$ time steps, the iterations converge and the final node representation $h_u^T$ can then be summarized with a readout(pooling) function $\text{R}(\circ)$ to extract graph-level representation $h_{\mathcal{G}}$.

$$h_{\mathcal{G}} = \text{R} \left( \left\{ h_v^T | v \subseteq \mathcal{V} \right\} \right). \quad (2)$$

There is a close connection between MPGNN and 1-WL test in that they both encode rooted subtrees, and is not general enough to capture arbitrary patterns in a graph. It is proved that MPGNN's discriminative power is upper bounded by 1-WL test, and hence cannot distinguish any two $n$-node $r$-regular non-isomorphic graphs. Despite subgraph GNN (see definition in the following section) is proposed recently, the running time complexity and memory consumption overhead is still an issue for large graphs.

## 3 PROPOSED METHOD

In this section, we propose our approach, **M**ulti-**H**op **M**ulti-**C**olor GNN (M$^2$HC) which also encodes a rooted subtree structure as 1-WL MPGNNs do. However, by incorporating multi-hop information with different colors assigned to $k$-hop neighbors $\mathcal{N}_k(v)$ of the root node $v$, M$^2$HC provably go beyond 1-WL discriminative power using only 1-layer message passing, therefore is both fast and memory efficient. The overall procedure of GNN-M$^2$HC-RW is illustrated in figure 1.

**Definition 1.** (*Rooted Subgraph*) Given a graph $\mathcal{G}$ and a node $v$, the height-$h$ rooted subgraph $\mathcal{G}_v^h$ is induced by the nodes $N_{\leq k}(v)$, i.e. the nodes whose distance to node $v$ is no greater than $k$.

The concept of *Rooted Subgraph* follows Zhang & Li (2021). In Zhang & Li (2021), instead of iteratively refining node $v$'s representation using Eq. 1, they propose a subgraph-based nested scheme to enhance the expressivity of MPGNNs. Similarly, Zhao et al. (2021) also proposes a rooted subgraph based approach named Subgraph-1-WL$^*$. The algorithm in (Zhang & Li, 2021) only runs for 1 iteration, while the subgraph-1-WL$^*$ runs for several iterations just as the standard 1-WL GNNs. We refer to these variants of neural WL test as **subgraph GNN**.

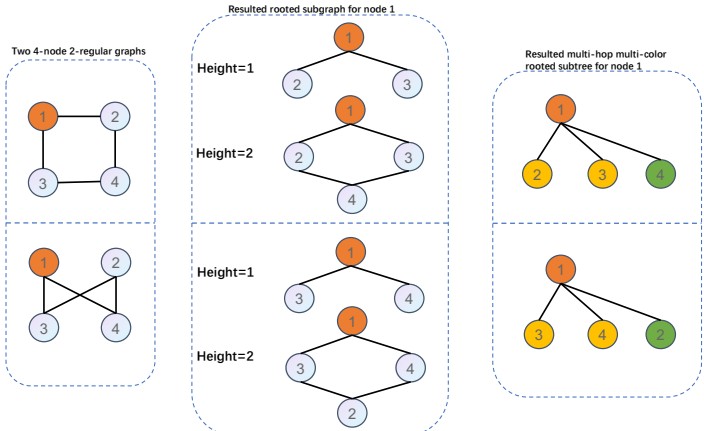

Figure 2: An illustration of resulted rooted subgraph and multi-hop multi-color rooted subtree for two non-isomorphic 4-node 2-regular graphs where the root node is node 1. As shown in the figure, both height-1 and height-2 rooted subgraph lead to the same representation for node 1. However using a 2-hop 2-color rooted subtree, node 1 receive different representations hence can be distinguishable.

As a rooted subgraph is extracted for every node $v \subseteq \mathcal{V}$ independently, a node $v$ in a different context (rooted subgraph induced by different nodes in the graph $\mathcal{G}$) may receive different representations, which is different from standard MPGNN in that a node only receive a universal representation. The discriminative power of subgraph GNN is enhanced due to the ability of each node being able to be aware of its context, i.e. the rooted subgraph induced by different nodes in the graph. It is guaranteed that subgraph GNN is able to exceed the expressive power of 1&2-WL test Zhao et al. (2021); Zhang & Li (2021).

**Definition 2.** (*Multi-hop multi-color rooted subtree*) For a $K$-hop rooted subgraph $\mathcal{G}_v^K$, $k$-hop neighborhood $\mathcal{N}_k(v)$ in $\mathcal{G}_v^K$ is extracted and directly connected to the root node $v$ and is assigned with a unique color $c_k$, where $k \in [1, K]$.

Figure 2 provides an example to illustrates the resulting *rooted subgraph* and ***Multi-Hop Multi-Color rooted subtree***(M²HC) for two 4-node 2-regular non-isormorphic graphs. Although the encoded rooted subgraph cannot distinguish the two graphs, M²HC is able to discriminate the structural disparity even if the node features remain the same. Hence, instead of using direct neighbors in WL test, we propose to leverage M²HC as a variant that could lead to the following message passing and update function:

$$
\begin{aligned}
m_v^t &= \text{AGG}^{(t)}\left(\left\{f_k\left(h_u^{t-1} \mid k \in [1, K], v \in V, u \in \hat{A}_k[v]\right)\right\}\right), \\
h_v^t &= \text{UPDATE}^{(t)}\left(h_v^{t-1}, m_v^t\right).
\end{aligned}
\tag{3}
$$

Here, $f_k(\circ)$ is the coloring function for the $k$-hop neighborhoods and $u$ is a $k$-hop neighbor of node $v$. By directly transferring M²HC into message passing function and run iteratively, the model expressivity will get uplifted, however we aim for a memory efficient and fast approach. As we will later prove that 1-iteration of M²HC is sufficient to go beyond 1-WL expressive power. Our message passing function for a $K$-hop M²HC is therefore simplified as:

$$
H = g\left(\left\{f_k\left(\hat{A}_k X\right) \mid k \in [1, K]\right\}\right),
\tag{4}
$$

where $g(\circ)$ is the node readout function to summarize information from $k \in [1, K]$ hop neighborhoods to get node representations $H \in R^{N \times F}$, where $F$ is the hidden dimensions of node $v \in \mathcal{V}$. The graph representation $R(H)$ can be further obtained with a readout function $R$. In all the experiments, we set $g(\circ)$ and $R(\circ)$ to be *SUM* or *MEAN* operators. Although being an extended version of *rooted subtree*, M²HC resembles subgraph GNN in that, thanks to the introduction of coloring function $f_k(\circ)$, the same node $v$ can also obtain a different representation for a different root node as the relative distance to a different root node is subject to change, and hence by exploiting $\hat{A}_k$, the subgraph structure is implicitly captured. Although being more expressive than rooted subtree structure, M²HC is less discriminative than subgraph GNN. In figure 3, we illustrate a failure case where subgraph GNN is able to capture the structural difference of the two non-isomorphic graphs

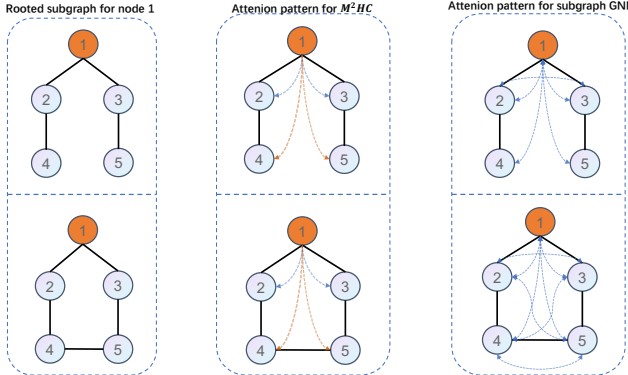

Figure 3: A failure case where subgraph GNN can distinguish the structrual difference while M$^2$HC cannot differntiate it. The figure also illustrate the attentional pattern induced by subgraph GNN and M$^2$HC, blue and orange dotted dash line denote different coloring functions. As shown in the figure, subgraph GNN and M$^2$HC lead to different attention patterns.

while M$^2$HC will output the same node representation and fail to discriminate them. In the following discussion, we treat message passing functions as a particular form of attention model, and provide an intuitive explanation of the expressive power possessed by M$^2$HC and subgraph GNN. As shown in figure 3, a node $v$ can attend to its $K$-hop neighboring nodes, and hence for subgraph GNN, it follows a bidirectional pairwise attention pattern while M$^2$HC follows a uni-directional hub-spoke pattern, as all the nodes in the subgraph can capture its local substructure, therefore with a injective *readout* function, subgraph GNN possess stronger expressive power than M$^2$HC.

To mitigate this weakness, we leverage scalable node feature augmentation method to make every node to be aware of its local substructure. We may resort to random features or one-hot features such as those in (Sato et al., 2021; Loukas, 2019). Although being simple and fast, adding random features would hurt the generalization performance, while one-hot features only adapt to transductive setting. Inspired by previous works on positional encoding (Dwivedi et al., 2021; You et al., 2019), we adopt $L$-step random walk self-landing probability of node $v$ to summarise $v$'s local graph structure as random walk can capture the status of a given node in the network. For instance, a hub node in the network will reach to itself much often than other nodes in a graph. Furthermore, A local change in the connection pattern will influence random walk dynamics, leading to a different pattern of self-landing statistics (Lovász, 1993). We denote $R = D^{-1}A$ as the random walk matrix, the $L$-step self landing probability vector for node $v$ equals to $(R_{vv}, R_{vv}^2, \ldots R_{vv}^L) \subseteq \mathcal{R}^L$. Let RW denote the self-landing probability matrix, the message passing function in matrix form for M$^2$HC is:

$$H = g\left(\left\{f_k\left(\hat{A}_k X' \mid k \in [1, K]\right)\right\}\right), \text{ where } X' = \text{CONCAT}(X, RW) \tag{5}$$

We term the model applying Eq. 5 as M$^2$HC-RW, and in all the experiments $f_k(\circ)$ is set to be 2-layer or 3-layer MLP due to the universal approximation theorem (Hornik et al., 1989; Hornik, 1991) and $g(\circ)$ to be *SUM* or *MEAN* pooling functions.

**Efficient Implementation.** Here we propose an efficient implementation for $\hat{A}_k$ as the main complexity of Eq. 5 stems from it.

**Fact.** The $(i, j)^{th}$ entry of $A^k$ counts the number of walk of length k where the start and end vertices are $i$ and $j$ respectively.

**Lemma 1** Let $d_{ij}$ be the distance between nodes $i$ and $j$. Then there exists a walk of length $d_{ij} + 2c$ between $i$ and $j$ for a simple graph where $c \in \{0, 1, 2, 3 \ldots\}$.

Using Lemma 1, we can come to the conclusion that $A^k \neq \hat{A}_k$ as the non-zero entries $\hat{A}_k[i, j] = 1$, s.t. $d_{ij} = k$ while $A^k[i, j] = d_{ij} + 2c$. However, we can calculate $\hat{A}_k$ using $A^k$ and $\left(\hat{A}_{k-1}, \hat{A}_{k-2}, \ldots, \hat{A}_1\right)$ efficiently using Eq. 6.

$$M_{k-1} = \gamma\left(\sum_1^{k-1} \hat{A}_j\right),$$

$$\left[A^k\right]_{ij} = 0, \forall (i, j) \in M_{k-1},$$

$$\widehat{A}_k = 1 \left\{ A_{ij}^k \geq 1 | i \times j \in |V| \times |V| \right\} . \tag{6}$$

Here, $\sum_1^{k-1} \hat{A}_j$ is a binary matrix where non-zero entries $(v, w)$ implies that $1 <= d_{vw} <= k - 1$, and $\gamma(\circ)$ returns a list of all the non-zero entries for $\sum_1^{k-1} \hat{A}_j$. The second line of the equation sets $\left[ A^k \right]_{ij}$ to zero where the indices $(i, j)$ is resulted from $M_{k-1}$, and therefore now $A^k$ only contains the non-zero elements $i, j$ where $d_{ij} = k$. The last line in Eq.6 outputs $\hat{A}_k$ as the binarized version of $A^k$, where $\left[ \hat{A}_k \right]_{ij} = 1, \forall d_{ij} = k$. The iterative realization of $\hat{A}_k$ is efficient as the computation only involves sparse matrix multiplication, addition and fancy indexing.

## 4 DISCUSSIONS

In this section, we theoretically analyze the expressivity of our proposed method, followed by a discussion of the computational complexity and memory requirement of M²HC-RW, finally We compare and contrast our method with other related works.

### 4.1 THEORETICAL ANALYSIS

In this subsection, we theoretically analyze the expressive power of M²HC-RW.

**Theorem 1** *Subgraph GNN is strictly more powerful than 1&2-WL.*

Theorem 1 can be directly obtained from *Theorem 2* of Zhao et al. (2021) and *Theorem 1* from Zhang & Li (2021). Although the proof sketch is not the quite the same, they both prove that subgraph GNN is more discriminative than 1-WL graph isormorphism test. As the discriminative power of 2-WL is equivalent to 1-WLMaron et al. (2019), subgraph GNN is proved to be more powerful than 1&2-WL. Next, we establish the connection between M²HC-RW and subgraph GNN in terms of their expressive power.

**Theorem 2** *M²HC-RW is at least as equivalent to the expressive power of subgraph GNN if the following conditions hold:*

1. *The coloring function $f(\circ)$,node readout function $g(\circ)$ and graph readout function $R$ in Eq. 5 should be injective.*

2. *For a K-hop M²HC and L-step random walk probability matrix RW, L should be sufficiently large to capture the structural information.*

A direct corollary derived from theorem 1 and 2 is:

**Corollary 2.1** *M²HC-RW is strictly more powerful than 1&2-WL.*

Theorem 2 is proved by observing that incorporating self-landing probabilities to augment node features could help resolve the expressivity issues incurred by hub-spoke attention pattern of M²HC. We also provide a toy example to illustrate that M²HC can successfully discriminate some cases where subgraph GNN fails to detect. Due to the exponential growth of K-hop neighbors, K should not be set too large, which might require more expressive model to extract meaningful representations Loukas (2019), however K should not be too small neither as it will lose useful information. $K$ is typically set to 3 or 4 in the experiments to strike a balance between model complexity and predictive performance.

**Theorem 3** *Let $P = \{0, 1\}^{n \times n}$ be any permutation matrix,if we define $\mathcal{T}$ to be an alias operator for M²HC-RW,then $\mathcal{T}(A, X) = \mathcal{T}\left(PAP^T, PX\right)$,i.e. M²HC-RW is permutation invariant.*

As M²HC-RW is experimented under the setting of graph classification, the permutation invariance property guarantees that for any node permutation $P$, the graph representation remains the same, hence is positive for the graph classification task.

## 4.2 COMPLEXITY ANALYSIS

As there are two stages for our approach, we analyze the running time complexity separately in the this section.

1. **Preprocessing.** In this stage, $\hat{A}_k$ and the self-landing probability matrix RW require some pre-computation. However, both of them can be computed efficiently. First, to compute $\hat{A}_k$, we only need to compute $A^k = A * A^{(k-1)}$, hence a bottom-up dynamic programming approach only lead to $\mathcal{O}(K * m)$ computational complexity for a K-hop $M^2$HC-RW and $m$ is the number of non-zero entries for adjacency matrix $A$. Similarly,we need to precompute $R, R^2, R^3, \ldots, R^L$ for a L-step RW probability matrix, we can use a similar approach leading to $\mathcal{O}(L * m)$ running time complexity. Lastly, we can also precompute $\hat{A}_k X$, for a sparse matrix $\mathcal{G}$, the time complexity is $O(mF)$ where $F$ is the feature dimension of $X$. For a sparse graph, as $K << m, L << m$ and $F << m$, the time complexity of the preprocessing stage is $\mathcal{O}(|V|)$.

2. **Training and Inference.** The computation for training and inference is fast and memory-efficient once the preprocessing stage is finished. As $f_k(\circ)$ in Eq. 5 is a 2-layer or 3-layer MLP, the computational complexity is $\mathcal{O}(|B|FH)$, where $|B|$ is the batch size, $F$ denotes the feature dimensions and $H$ denotes the hidden dimensions of $f_k(\circ)$. $M^2$HC-RW can be even faster than 1-WL MPGNNs while being more expressive. Notably for Nested GNN Zhang & Li (2021), the computational complexity is $\mathcal{O}(|B|hdFH)$, where $h$ is the height of extracted rooted subgraph which is typically 4 or 5, and $d$ is the average node degree. As subgraph GNN incurs an additional memory overhead of $hd$, $M^2$HC-RW can consume **10x-50x less memory** than subgraph GNN assuming $d$ ranges between 3 to 10.

## 4.3 RELATED WORK

Detailed discussion on the related works is shown in appendix A.3.

## 5 EXPERIMENTS

The following research questions guide the remainder of the paper: **(Q1)** Can $M^2$HC-RW go beyond 1-WL expressive power? **(Q2)** Is $M^2$HC-RW able to outperform 1-WL MPGNNs in real-world datasets? **(Q3)** How does $M^2$HC-RW perform in large-scale benchmark graph datasets? **(Q4)** How much training time and memory consumption do $M^2$HC-RW incur?

We make use of Pytorch Geometric (Fey & Lenssen, 2019) to implement our proposed framework, and Pytorch Lightning that provides higher-level abstraction built upon PyTorch (Paszke et al., 2019) for efficient model training and inference. Our code is available at `https://github.com/reywqua/ICLR2023_2923`.

## 5.1 DATASETS

To answer **Q1**, we make use of the following synthetic datasets to examine if $M^2$HC-RW can exceed 1&2-WL expressive power. *i) LCC(X)* Sato et al. (2021), which contains 20-node 3-regular random graphs for multi-label node classification problem, the class label for node $v$ is the local clustering coefficient of node $v$. Both training set and test set consist of 1,000 graphs. *ii) TRI(X)* Sato et al. (2021) which also consists of 2,000 20-node 3-regular random graphs split by half for training and testing, the goal is to predict if two neighboring nodes are adjacent to each other. *iii) EXP* Abboud et al. (2020) contains 600 pairs of 1&2-WL failed graphs that are split into two classes where each graph of a pair is assigned to two different classes. *iv)* Graphlet counting Abboud et al. (2020). The goal is to count four substructures's number in random graphs,namely 3-star, traingle, tailed-triangle and 4-cycle. For TRI(X) and LCC(X) datasets, the evaluation metric is ROC-AUC as the label distribution is skewed. For EXP dataset, the evaluation metric is Accuracy, and for graphlet counting dataset, MAE is adopted. To answer **Q2**, We adopt ENZYMES, DD, PROTEINS, MUTAG, NCI1 and BZR from TUDataset Morris et al. (2020) which is self-contained in Pytorch Geometric Fey & Lenssen (2019). We adopt Accuracy as the evaluation metric. To answer **Q3**, *ogb-molhiv* and *ogb-molpcba* from Open Graph Benchmark datasets Hu et al. (2020) and *zinc-12k* are used to verify the

Table 2: Results on counting 4 different substructures, the evaluation metric is Mean Square Error. Truncation is performed when test error drops below 1.0E-4.

| Method | 3-Star | Triangle | Tailed_Tri. | 4-Cycle |
|---|---|---|---|---|
| GCN | 1.0E-4 | 2.43E-1 | 1.42E-1 | 1.14E-1 |
| GAT | 1.0E-4 | 2.47E-1 | 1.44E-1 | 1.12E-1 |
| GIN | 1.0E-4 | 2.06E-1 | 1.18E-1 | 1.21E-1 |
| PPGN | 1.0E-4 | 1.0E-4 | 2.61E-4 | 3.30E-4 |
| $M^2$HC | 1.0E-4 | 8.61E-4 | 2.20E-3 | 5.83E-3 |
| GIN-$M^2$HC-RW | 1.0E-4 | 8.76E-4 | 2.25E-3 | 5.27E-3 |
| PPGN-$M^2$HC-RW | **1.0E-4** | **1.0E-4** | **1.0E-4** | **1.0E-4** |

expressive power of the proposed method. The ogb-molhiv dataset contains 41K small molecules, the task of which is to classify whether a molecule exhibits HIV virus or not. *ROC-AUC* is the standard evaluation metric for this dataset. The ogbg-molpcba dataset contains 438K molecules with 128 classification tasks. The evaluation metric is *Average Precision* over all the classification tasks. zinc-12k (Sterling & Irwin, 2015; Gómez-Bombarelli et al., 2018; Dwivedi et al., 2020) dataset contains 12K molecule graphs to regress a molecular property known as constrained solubility. Following Dwivedi et al. (2020) we adopt Mean Absolute Error(MAE) as evaluation metric.

## 5.2 EXPERIMENT RESULTS

For **Q1**, as illustrated in table 1, $M^2$HC-RW is able to achieve 99.7% and 1.0 in LCC(X) and TRI(X) respectively, where GIN and GCN can only achieve 50% in terms of ROC-AUC score. For EXP dataset, $M^2$HC-RW achieves 99.8% accuracy while 1-WL GNNs only achieves 50%. This implies that $M^2$HC-RW exceeds 1-WL discriminative power and can discriminate the substructures that GIN and GCN do no better than random guess. For the graphlet counting dataset, the experiment result is shown in table 2. $M^2$HC-RW achieves much lower MAE compared with 1-WL GNNs. Second, equipped with $M^2$HC-RW, GIN and PPGN Maron et al. (2019) demonstrate higher expressive power, i.e., the resulting MAE is significantly less than the models without $M^2$HC-

Table 1: Results on LCC(X),TRI(X) and EXP. ROC-AUC is adopted as the evaluation metric for LCC(X) and TRI(X), and Accuracy(%) is used for EXP.

| Method | LCC(X) | TRI(X) | EXP |
|---|---|---|---|
| GCN | 50% | 50% | 50% |
| GIN | 50% | 50% | 50% |
| $M^2$HC-RW | 99.7% | 100% | 99.8% |

RW. This also implies that $M^2$HC-RW can be equipped with more advanced models such as PPGN, to uplift their expressivity. Finally, PPGN with $M^2$HC-RW achieves the best MAE error consistently in all of the four substructures.

To answer **Q2**, first we can see that a 1-layer $M^2$HC-RW can outperform all the 1-WL GNNs in almost all the datasets as shown in table 3, in most cases by a large margin. We also test GNN-RW to verify the improved expressive power doesn't come from injecting random-walk positional features-although the accuracy does improve a little bit in some datasets, GNN-$M^2$HC-RW improves much more significantly. Finally 1-WL GNNs, together with $M^2$HC-RW uplifts the model expressivity in almost all datasets. This implies that by encoding a rooted subtree and multi-hop multi-color rooted subtree in the message passing procedure, the model can leverage both inductive bias to extract more meaningful representations. For all the datasets except MUTAG, the best performing model is achieved by GNN-$M^2$HC-RW or $M^2$HC-RW.

To answer **Q3**, we adopt GIN+Virtual_Node(GIN$^*$) as the base GNN method in GNN-$M^2$HC-RW for ogbg-molhiv and ogbg-molpcba, we can see that by equipping with $M^2$HC-RW module, GIN$^*$'s discriminative power has been improved significantly and consistently in both datasets- from 77.07% to 78.21% in ogb-molhiv and 26.86% to 27.92% in ogbg-molpcba as illustrated in table 5. GIN$^*$-$M^2$HC-RW is even comparable with or outperform some heavily-tuned leading methods. Finally, our method is comparable with subgraph GNN methods, i.e., Nested GNN Zhang & Li (2021) and GNN-AK Zhao et al. (2021), which is also consistent with our theorem in terms of model expressivity. Finally we test GNN and GNN-$M^2$HC-RW on zinc-12k dataset which is a large scale molecular benchmark. We use GCN,GIN,GraphSage and PPGN as the baseline methods and use only node label features, without leveraging any edge features to verify whether GNN-$M^2$HC-RW can outperform the base GNN methods. As demonstrated in table 6, GNN-$M^2$HC-RW outperforms

Table 3: Graph classification results on TU datasets in terms of Accuracy(%) and one standard deviation, we use **bold** to highlight the best performing method.

| | D&D | PROTEINS | MUTAG | NCI1 | BZR | ENZYMES |
|---|---|---|---|---|---|---|
| GCN | 68.64±4.7 | 69.42±4.1 | 74.73±6.8 | 64.92±1.1 | 80.93±5.1 | 24.21±3.7 |
| GCN-RW | 69.24±2.1 | 69.47±3.4 | 70.72±9.1 | 65.7±2.2 | 82.85±4.4 | 25.29±6.0 |
| GCN-M$^2$HC-RW | **75.6±5.2** | 74.5±1.8 | 85.3±10.2 | 73.38±1.7 | 81.23±3.9 | 36.66±7.1 |
| Graphsage | 69.79±4.4 | 68.2±3.2 | 72.38±5.4 | 68.34±1.9 | 81.1±5.6 | 25.1±6.0 |
| Graphsage-RW | 70.12±3.5 | 69.6±3.1 | 75.98±5.8 | 68.74±1.8 | 82.77±3.6 | 23.33±6.2 |
| Graphsage-M$^2$HC-RW | 73.71±2.51 | 73.7±2.2 | 86.2±8.1 | 74.06±1.2 | 82.78±4.0 | 38.19±3.9 |
| GIN | 70.98±4.5 | 70.3±3.1 | **87.89±7.9** | 71.61±2.0 | 85.9±3.6 | 32.01±7.3 |
| GIN-RW | 70.85±1.5 | 69.54±3.0 | 85.84±6.2 | 71.8±1.4 | 86.32±4.4 | 31.66±4.7 |
| GIN-M$^2$HC-RW | 72.26±3.3 | **74.8±1.1** | 84.5±6.53 | 74.17±1.7 | **87.51±4.8** | **38.75±3.6** |
| M$^2$HC-RW | 73.41±3.8 | 73.22±2.6 | 84.99±8.6 | **75.29±1.1** | 83.61±4.4 | 34.54±5.8 |
| MAX IMPROVEMENT | 10.14% | 9.67% | 19.22% | 13.03% | 2.07% | 52.76% |

Table 4: Normalized running time and memory overhead for various methods. Our approach incurs significantly less memory usage.

| | Zinc-12k | |
|---|---|---|
| **Method** | **Run Time(S/Epoch)** | **Memory(MB)** |
| GIN | 1 | 1 |
| GIN-AK$^*$ | 1.57 | 15.41 |
| GIN-AK$^*$-S(R=5) | 2.08 | 15.008 |
| GIN-AK$^*$-S(R=3) | 2 | 11.23 |
| GIN-AK$^*$-S(R=1) | 1.8 | 3.16 |
| GIN-M$^2$HC-RW | **1.24** | **1.04** |

all four GNN variants. PPGN-M$^2$HC-RW achieves the best MAE across all the methods. Detailed experiment results are illustrated in appendix A.2.

To answer **Q4**, we compare GNN-M$^2$HC-RW with GIN-AK$^*$ on zinc-12k in terms of memory usage and per epoch running time. As the experiment setting is different, we measure normalized running time and memory overhead where the time and memory incurred by GIN equals to 1. The experiment result is shown in table 4. We should also note that GIN-AK$^*$Zhao et al. (2021) is different with GIN-AK in that GIN-AK$^*$ further leverages context encoding and distance-to-centroid encoding in each layer to improve the model capacity, however the memory cost mainly arises from independently rooted subgraph extraction for each node in the graph, hence the memory overhead would be similar to GNN-AK. As we can see in table 4, add M$^2$HC-RW on GIN is nearly memory-free, the additional memory consumption stems from the model parameters of coloring function $f_k(\circ)$ in Eq. 5. As $AX$ can be precomputed, the training and inference procedure is orthogonal to the graph structure information, hence is also fast. GIN-M$^2$HC-RW consumes only 1/15 memory overhead compared with GIN-AK$^*$. Although GIN-AK$^*$-S introduces sampling operation, the training becomes slower as the sampling operation introduces more computation complexity to avoid redundant rooted subgraphs, and the inference stage, without using subgraph sampling, is still memory inefficient.In terms of speed, GIN-M$^2$HC-RW is 43% faster than GIN-AK$^*$-S(R=1) and is 25.6% faster than GIN-AK$^*$.

# 6 CONCLUSION

In this paper we develop M$^2$HC-RW, a more generalized variant of neural Weisfeiler-Lehman test to uplift GNN's representative power. M$^2$HC-RW is fast, flexible and memory efficient. Furthermore, it is provably more expressive than 1-WL graph isomorphism testing using only 1 iteration of message passing. Compared with other more powerful GNNs, M$^2$HC-RW requires significantly less memory overhead, and can be applied to large-scale graphs. M$^2$HC-RW can also be deployed to low-resource device for training and inference due to its low computational complexity and memory efficiency. In the future, we seek to incorporate attention mechanism into M$^2$HC-RW and make it compatible with edge attributes.

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

# A    APPENDIX

## A.1    EXPERIMENT SETTING

For the synthetic datasets, we directly verify the model performance of M$^2$HC-RW on LCC(X), TRI(X) and EXP. The baseline methods are GCN Kipf & Welling (2016) and GIN Xu et al. (2018a). For graphlet counting dataset, GCN,GAT Veličković et al. (2017), Graphsage, PPGN are used as the baseline methods. For GIN and PPGN, M$^2$HC-RW is also equipped to each of these methods to verify if there is uplift for the model expressivity. Besides, M$^2$HC-RW is also experimented alone. All methods are trained for 100 epochs. The best performing model at validation stage with specified evaluation metric is used for test dataset.

For the TUDatasets, we adopt GCN, GraphSage Hamilton et al. (2017) and GIN as the base methods. To demonstrate the uplifted power is not solely stemming from random-walk positional features, we also evaluate the performance of GCN-RW, GraphSage-RW and GIN-RW. Two variants of our proposed method are experimented, namely M$^2$HC-RW and GNN-M$^2$HC-RW. GNN-M$^2$HC-RW leverages both M$^2$HC-RW and a base GNN method, and concatenates the graph-level representations from both models for the final classification. For all GNN models including GNN-M$^2$HC-RW, the message passing layers in $\{2, 3, 4, 5\}$ is searched, and the hidden dimensions of $\{16, 32\}$ is searched. For M$^2$HC-RW, *3*-hop with 10-step random walk is used uniformly in all the experiments without any hyperparameter search. All models are trained uniformly for 150 epochs with 32 batch-size. We adopt the 10-fold cross validation method to ensure a fair comparison. For each fold, we record the test accuracy according to the best validation performance and report the average accuracy across all 10 folds. It is also noteworthy that to ensure a fair comparison, Jumping Knowledge Xu et al. (2018b) is not incorporated in GIN and all other methods, which is leveraged in Xu et al. (2018a) to enhance GIN's model capacity.

We then turn to large scale datasets to verify the performance of GNN-M$^2$HC-RW and M$^2$HC-RW. For ogb-molhiv and ogbg-molpcba, we use GIN$^*$-M$^2$HC-RW, and compare with other advanced GNN methods, where GIN$^*$ refers to GIN+Virtual Node. A 1-layer or 2-layer K-hop L-step M$^2$HC-RW is used, where K is set to 3 and L equals to 20. For zinc-12k dataset, we ignore edge features and only leverage node features to facilitate the comparison between M$^2$HC-RW and other baseline methods, namely GCN, Graphsage, GIN and PPGN. Finally, we compare base GNN methods with and without M$^2$HC-RW module to verify if equipped with our proposed method can enhance the expressivity of base GNNs.

## A.2    EXPERIMENT RESULTS ON OGBG AND ZINC-12K DATASETS

Table 5 illustrates the experimental results for ogbg-molhiv and ogbg-molpcba. As we can see, by equipping with M$^2$HC-RW module, GIN$^*$'s discriminative power has been improved significantly and consistently in both datasets- from 77.07% to 78.21% in ogb-molhiv and 26.86% to 27.92% in ogbg-molpcba. Our approach is also comparable with subgraph GNN methods, i.e., Nested GNN Zhang & Li (2021) and GNN-AK Zhao et al. (2021), which is consistent with our theorem in terms of model expressivity.

Table 6 shows the experimental result for zinc-12k dataset. As demonstrated in the table, GNN-M$^2$HC-RW outperforms all four GNN variants. PPGN-M$^2$HC-RW achieves the best MAE across all the methods.

Table 5: Graph classification result on ogbg-molhiv and ogbg-molpcba in terms of *roc-auc* and *average precision*

| Method | ogbg-molhiv | ogbg-molpcba |
|---|---|---|
| | Test | Test |
| CCN | 75.99±1.19 | - |
| PNA | 79.05±1.32 | 28.38±0.35 |
| DGN | 79.70±0.97 | 28.85±0.3 |
| DeeperGCN | - | 27.81±0.38 |
| CIN | 80.94±0.57 | - |
| GIN | 77.07±1.49 | 27.03±0.23 |
| Nested-GIN | 78.34±1.86 | 28.32±0.41 |
| GIN-AK | 78.29±1.21 | 27.40±0.32 |
| GIN-M$^2$HC-RW | 78.21±1.03 | 27.92±0.12 |

Table 6: Test results for zinc-12k dataset. **bold** indicates the best performing model in the test dataset. Only node feature is leveraged, the edge feature is not leveraged even when it is available.

| Method | ZINC-12k(MAE) |
|---|---|
| GCN | 0.317±0.012 |
| GCN-M$^2$HC-RW | 0.26±0.004 |
| GIN | 0.316±0.003 |
| GIN-M$^2$HC-RW | 0.266±0.003 |
| SAGE | 0.269±0.004 |
| SAGE-M$^2$HC-RW | 0.247±0.004 |
| PPGN | 0.306±0.051 |
| PPGN-M$^2$HC-RW | **0.182±0.002** |

### A.3 RELATED WORK

In this section, we compare and contrast our approach with previous relevant works. As the expressivity of standard message passing GNNs, e.g. GCN and GIN is upper bounded by 1-WL graph isomorphism testing, a fruitful line of works has been proposed aligning to k-WL test. Although the expressivity of these methods get uplifted with theoretical guarantee, most of them lacks an important feature contributing to the recent success of GNNs: the locality of computations. Therefore, they hold more theoretical values than practical availability. However, our approach guarantees to surpass 1-WL expressivity without losing the benefit of localized computations.

In Zhang & Li (2021); Zhao et al. (2021), the authors propose a variant of 1-WL test: instead of hash the direct neighbors of a given node, a rooted subgraph is utilized for representation refinement. Although these approaches(subgraph GNNs) enjoy theoretical guarantee to surpass the expressivity of 1-WL test and preserves locality of computation, it requires expensive memory consumption as an independent rooted subgraph needs to be extracted for each node in the graph. Thanks to this operation, each node is able to receive different representations specific to the node-induced rooted subgraph, which is different to 1-WL GNNs where a node calculates a universal representation based on the rooted subtree. M$^2$HC inherits the advantage from subgraph GNNs that a node get a localized representation based on the root node $v$ and the distance to $v$. M$^2$HC also adopts the idea of subgraph pooling from subgraph GNNs, i.e. the representation of a root node is obtained by pooling the multi-hop multi-color rooted subtree instead of calculating a node representation in a layer-by-layer regime in 1-WL GNNs. Due to the adopt of $\hat{A}_k$ and precomputation of $\hat{A}_k X$, M$^2$HC is able to capture the notion of rooted subgraph implicitly without the need of rooted subgraph extraction, hence is significantly memory efficient and fast in both training and inference stage. Although M$^2$HC is less expressive than subgraph GNNs due to its hub-spoke attention pattern, by adding random-walk self-landing probability as auxiliary feature, M$^2$HC-RW is guaranteed to be as least expressive as subgraph GNNs. Another difference is that subgraph GNNs encode rooted subgraph in the neural WL test, while GNN-M$^2$HC-RW leverages both rooted subtree and rooted multi-hop multi-color subtree to calculate the node representation, hence GNN-M$^2$HC-RW can be

considered as an ensemble method where standard 1-WL test and a more generalized version of 1-WL test are both leveraged.

Recent works propose to provably enhance the expressivity of GNNs by augmenting node features with random feature or auxiliary coloringsSato et al. (2021); Loukas (2019); Abboud et al. (2020); Puny et al. (2020); Dasoulas et al. (2019), although being scalable and easy to implement, these approaches may not preserve permutation invariant property, thus would hurt the generalization performance outside of the training set. However, leveraging random-walk features as augmentation to M$^2$HC doesn't hurt the generalization performance as it is provably permutation invariant. Instead, ID-GNNYou et al. (2021) admits node coloring using different weight matrices, which is similar to the coloring function in our work to tag node colors. ID-GNN still uses standard 1-WL test, leading to a rooted subtree structure, the extra discriminative power stems from coloring the identity node. Meanwhile, we propose a new variant of 1-WL test by hashing a multi-hop multi-color rooted subtree. With a 1-layer message passing function, M$^2$HC-RW is guaranteed to go beyond 1-WL expressivity thanks to the localized node representation and pooling operation. Another line of works resort to graph positional encoding to break the limit of 1-WL expressive power by informing the localized substructure of arbitrary nodes in a graphDwivedi et al. (2020; 2021); You et al. (2019). unlike many previous works that inject positional encodings in 1-WL GNNs, we leverage positional encoding in our proposed multi-hop multi-color rooted subtree, leading to several advantages in uplifting the model expressivity: i) The view of each node is confined to rooted node $v$ in M$^2$HC, due to the hub-spoke attention flow, the internal connection pattern induced by non-root nodes are ignored which may lead to under-expressivity, the employee of graph positional encoding could capture the structural disparity therefore enrich the node representations. ii) Graph positional encoding is not restricted to a single M$^2$HC, and can capture expected statistics over all the M$^2$HC induced by its neighboring nodes, hence can be viewed as context encodings. iii) A node's representation conditions on both global-level positional encoding and local-level distance information (via coloring funtion $f_k(\circ)$), leading to higher-quality representations. iv) The random-walk features is permutation invariant, therefore won't hurt the generalization performance outside of training set.

Recent work also proposes to use distance features to uplift GNN's expressivityLi et al. (2020), where a distance vector w.r.t the target node set is calculated for each node as its additional feature. Our approach is compatible with distance-based features due to the adoption of coloring function $f_K(\circ)$. Similar to our framework, Nikolentzos et al. (2020); Abu-El-Haija et al. (2019) also leverages multi-hop neighborhood to perform higher-order message passing. However, MixHopAbu-El-Haija et al. (2019) directly utilizes $A^k$ in the message passing function, resulting in the entanglement of information from multi-hop neighborhood to the central node. While our framework leverages $\hat{A}_k$ to disentangle the information from multi-hop neighborhoods to the root node, leading to a hub-spoke multi-color attention pattern. Nikolentzos et al. (2020) directly utilizes k-hop neighbors in the aggregation function instead of direct neighbors. Both Nikolentzos et al. (2020); Abu-El-Haija et al. (2019) incorporate multi-hop neighborhood information into the message passing function and run for several iterations. However, M$^2$HC extends the rooted subtree in WL test and is provably more expressive than 1-WL GNNs with only 1 iteration.

## A.4 PROOF OF THEOREM 2

We consider two graphs $\mathcal{G}^1$ and $\mathcal{G}^2$, and select two nodes $v_1 \in \mathcal{G}^1$ and $v_2 \in \mathcal{G}^2$. For the extracted height-$h$ rooted subgraph $\mathcal{G}^h_{v_1}$ and $\mathcal{G}^h_{v_2}$, let $Q^h_{v_1}$ denotes the number of nodes in $\mathcal{G}^h_{v_1}$ whose distance to root node $v_1$ equals to $h$, similarly for $Q^h_{v_2}$. We consider the first case that $\{Q^1_{v_1}, Q^2_{v_1}, \ldots Q^h_{v_1}\} \neq \{Q^1_{v_2}, Q^2_{v_2}, \ldots Q^h_{v_2}\}$, which means that at least one of the element $Q^k_{v_1} \neq Q^k_{v_2}$, as the topology of the two rooted subgraphs $\mathcal{G}^h_{v_1}$ and $\mathcal{G}^h_{v_2}$ is not the same, the resulted representation from subgraph GNN for $v_1$ and $v_2$ will not be the same. Similarly, as the coloring function and readout function in M$^2$HC is injective, the node representation for $v_1$ and $v_2$ will also be different. Consider another case that $\{Q^1_{v_1}, Q^2_{v_1}, \ldots Q^h_{v_1}\} = \{Q^1_{v_2}, Q^2_{v_2}, \ldots Q^h_{v_2}\}$,however the internal structure for $\mathcal{G}^h_{v_1}$ and $\mathcal{G}^h_{v_2}$ is different, figure 3 illustrates one such scenario for $\mathcal{G}^h_{v_1}$ and $\mathcal{G}^h_{v_2}$ where $h = 2$. Due to the bidirectional pairwise attention, subgraph GNN is able to capture the difference and generate different representations for $v_1$ and $v_2$, yet M$^2$HC cannot distinguish the difference due to its unidirectional hub-spoke attention. However, with random walk features of sufficient steps, M$^2$HC-RW can still be aware of the structural disparity thanks to the node feature augmentation. Finally, we consider the case where $\mathcal{G}^h_{v_1}$ is structurally identical to $\mathcal{G}^h_{v_2}$, hence for subgraph GNN the representation for $v_1$ and

$v_2$ is identical, but is different for M$^2$HC-RW, as by incorporating random walk features every node in the graph can be aware of global structural disparity. Figure 4 demonstrates one such scenario where $v_1$ and $v_2$ leads to the same rooted subgraph hence the same representation using subgraph GNN while receiving different representation using M$^2$HC-RW. This also stresses the advantage of our approach that the receptive field of each node $v \in \mathcal{G}$ is enlarged given sufficient steps of random walk.

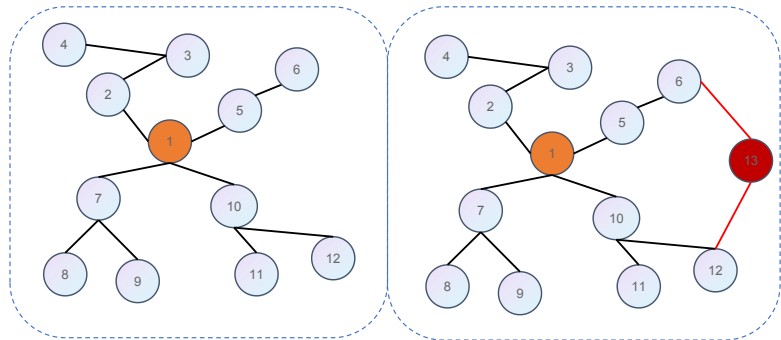

Figure 4: One scenario where node 1 in $\mathcal{G}^1$ and $\mathcal{G}^2$ is able to capture the structural disparity with M$^2$HC-RW, but not with subgraph GNN as the resulted height-2 rooted subgraphs for node 1 in the two figures are identical.

## A.5 PROOF OF THEOREM 3

Let R=$AD^{-1}$ denote the random walk matrix and $RW = \left[\text{diag}(R), \text{diag}\left(R^2\right), \ldots, \text{diag}\left(R^L\right)\right]$ be the L-step self-landing probability matrix. $diag(\circ)$ is an operator that takes in a matrix and return a diagonal column vector. Given an arbitrary permutation matrix $P = \{0, 1\}^{n \times n}$, we need to prove that $\mathcal{T}(A, X) = \mathcal{T}\left(PAP^T, PX\right)$, where $\mathcal{T}$ is the M$^2$HC-RW operator and X is the input that consists of node features and self-landing random walk features RW. We expand $\mathcal{T}$ and get:

$$
\begin{aligned}
H &= g\left(\left\{f_k\left(\hat{A}_k[X, RW]\right)\right\}\right) \\
&= g\left(\left\{f_k\left(\left[\hat{A}_k X, \left[\text{diag}\left(r^{-1}\hat{A}_k A\right), \text{diag}\left(r^{-2}\hat{A}_k A^2\right), \ldots, \text{diag}\left(r^{-L}\hat{A}_k A^L\right)\right]\right]\right)\right\}\right)
\end{aligned}
\tag{7}
$$

In Eq. 7 we abuse $X$ to be the node feature matrix, the input is a concatenation of node feature matrix X and L-step self-landing probability matrix RW. $K$ denote the height of the M$^2$HC-RW, i.e. K-hop M$^2$HC-RW. For simplicity, we assume a r-regular graph, hence $AD^{-1} = r^{-1}A$. With a transformation of a permutation matrix P, we have:

$$
\begin{aligned}
H_\pi &= g\left(\left\{f_k\left(P\hat{A}_k P^T\left[PX, PRWP^T\right]\right)\right\}\right) \\
&= g\left(\left\{f_k\left(\left[P\hat{A}_k X, P\hat{A}_k\left[\text{diag}\left(r^{-1}P^T PAP^T\right), \text{diag}\left(r^{-2}P^T PA^2 P^T\right), \ldots, \text{diag}\left(r^{-L}P^T PA^L P^T\right)\right]\right]\right)\right\}\right) \\
&= g\left(\left\{f_k\left(\left[P\hat{A}_k X, P\hat{A}_k\left[\text{diag}\left(r^{-1}AP^T\right), \text{diag}\left(r^{-2}A^2 P^T\right), \ldots, \text{diag}\left(r^{-L}A^L P^T\right)\right]\right]\right)\right\}\right) \\
&= g\left(\left\{f_k\left(\left[P\hat{A}_k X, \left[\text{diag}\left(r^{-1}P\hat{A}_k AP^T\right), \text{diag}\left(r^{-2}P\hat{A}_k A^2 P^T\right), \ldots, \text{diag}\left(r^{-L}P\hat{A}_k A^L P^T\right)\right]\right]\right)\right\}\right)
\end{aligned}
\tag{8}
$$

Clearly, $P\hat{A}_k X$ is a row-permutation of $\hat{A}_k X$, with a injective function $f(\circ)$ and a injective readout function $g(\circ)$, $g\left(\left\{f_k\left(\hat{A}_k X\right)\right\}\right) = g\left(\left\{f_k\left(P\hat{A}_k X\right)\right\}\right)$ holds true. Next, we compare $\text{diag}\left(r^{-l}\hat{A}_k A^l\right)$ with $\text{diag}\left(r^{-l}P\hat{A}_k A^l P^T\right)$.

As $\text{diag}\left(P\hat{A}_k A^l P^T\right)$ return a permutation of $\text{diag}\left(\hat{A}_k A^l\right)$ without changing element values, and this holds true for all the $l \in [1, L]$, hence $PRWP^T$ returns a row-permutation of RW, and per-

mutation invariance property also holds for RW. Based on the two observations, we come to the conclusion that $M^2$HC-RW is permutation invariant.

