# OpenReview forum: "GOING BEYOND 1-WL EXPRESSIVE POWER WITH 1-LAYER GRAPH NEURAL NETWORKS"
_ICLR.cc/2023/Conference — Submitted to ICLR 2023_

### Official Review · Reviewer_xS7k · 2022-10-16

**Confidence:** 4
**Correctness:** 3
**Technical Novelty And Significance:** 3
**Empirical Novelty And Significance:** Not applicable
**Recommendation:** 3

**Clarity, Quality, Novelty And Reproducibility:**

Novelty:
The algorithm proposed tries to find a balance between expressive power and scalability, which is a good attempt. However, the method itself is more like a combination of some already existing theories and methods.

Quality:
No critical flaws are identified. The theoretical analysis doesn't contain much to check though.

Clarity:
The paper contains a few sentences that are too long, which hinder understanding. Other than that, the paper is quite clear.

Reproducibility:
The code is provided. The discussion on experiment setups is clear.

Question:


**Strength And Weaknesses:**

Strength:
1. Reduce the memory requirement by compressing the multi-layer structure of the message passing GNNs into one iteration.
2. The idea of the multi-hop multi-color subtree is straightforward and easy to understand.

Weaknesses:
1. The theoretical analysis is a mostly intuitive discussion with no mathematical proof.
2. In the method section, too much effort is spent on discussing the subgraph GNNs.
3. The datasets in the experiments for Q1 are questionable. For a node classification problem, the training set and the test set are 1000 graphs each, indicating that the model trained is transferred from the training graph to the testing graphs. While for a node classification task, the norm is training on some nodes of a graph and testing on the other nodes of the same graph.
4. The paper claims that the method is faster and more memory efficient. Memory efficiency is indeed significant, while the run time doesn't improve significantly in the experiment, and hasn't been discussed theoretically.

**Summary Of The Paper:**

The paper proposes a light-weighted plugin/algorithm to uplift the upper bound of the GNNs' expressive power, namely the 1-WL test. By constructing a multi-hop multi-color rooted subtree, the algorithm achieves or even exceeds the performance of the subgraph GNNs while using fewer computational resources.

**Summary Of The Review:**

The paper presents an interesting alternative compared to the message-passing GNNs and the subgraph GNNs. However, the novelty is limited and the theoretical discussion is a little weak. Also, there are minor flaws in the experiment.

---

### Official Review · Reviewer_w5t6 · 2022-10-18

**Confidence:** 4
**Correctness:** 2
**Technical Novelty And Significance:** 2
**Empirical Novelty And Significance:** 2
**Recommendation:** 3

**Clarity, Quality, Novelty And Reproducibility:**

**Clarity**
Although the presented idea is simple it lacks clarity, especially with regard to mathematical rigor. Some examples:
- Section 2.1, the definition of $\hat{A}(i)$ is unclear to me
- Section 2.2, the WL test should be formally defined as it is central to the paper
- Section 3, "subgraph GNN" is only informally introduced although a formal definition is needed to check the proofs
- Eq. 3, the scope of the variable $k$ is unclear.  The equation needs more explanation
- Lemma 3, the statement is unclear and does not seem to be correct in general
- Theorem 1, seems to be taken from the literature without proper reference
- Theorem 2, the statement is not precise, e.g., what does "sufficiently large" mean? Further, the proof is too handwavey to verify its correctness.
- $\dots$



**Comments**
- Theorem 3 appears out of nowhere
- Nowadays there are a lot more papers using subgraph information, e.g.,
   - Leonardo Cotta, Christopher Morris, Bruno Ribeiro: Reconstruction for Powerful Graph Representations. NeurIPS 2021.
  - Beatrice Bevilacqua, Fabrizio Frasca, Derek Lim, Balasubramaniam Srinivasan, Chen Cai, Gopinath Balamurugan, Michael M. Bronstein, Haggai Maron: Equivariant Subgraph Aggregation Networks. ICLR 2022
  - $\dots$

**Minor points**
- The paper suffers from many typos and grammatical errors. Thorough proofreading is needed. For example in the abstract:
   - "Weisfeiler-Lehman Test" -> "Weisfeiler-Lehman test"
   - "It is shown theoretically our method" -> "It is shown theoretically that our method"
   - "The Numerical" -> "The numerical"
   - $\dots$

**Strength And Weaknesses:**

**Strong**
- Simple approach that sometimes leads to an empirical boost

**Weak**
- Presentation and formalization is subpar
- Proofs are of a hand-wavey nature and not formal enough to verify.
- Related work section misses many papers


**Summary Of The Paper:**

The paper deals with supervised machine learning for graphs via GNNs. Specifically, the authors propose a simple way of going beyond GNNs' limits in expressivity ($1$-WL). That is, during aggregation, the authors propose to not only consider direct neighbors but also neighbors $k$-hops away. To distinguish these from direct neighbors they are given unique labelings (for each $k$). To further enhance the expressivity the authors concatenate random walk self-landing probabiliies to the node features, e.g., following Dwivedi et al., 2021.





**Summary Of The Review:**

This is an incremental paper. In its current form of presentation and its lack of formalization, it is not ready for a top-tier conference.

---

### Official Review · Reviewer_8LVG · 2022-10-24

**Confidence:** 4
**Clarity, Quality, Novelty And Reproducibility:** The paper is clearly written.
**Correctness:** 3
**Technical Novelty And Significance:** 2
**Empirical Novelty And Significance:** 2
**Recommendation:** 3

**Strength And Weaknesses:**

Strengths:

1. The motivation of encoding multi-hop subtree with multiple colors is technically sound.
2. This manuscript is clearly written and organized.

Weakness:

1. The novelty of the proposed method is limited. This method simply encoding multi-hop subtree to improve the expressiveness of GNNs. However, this idea has been extensively used by previous works, such as SIGN [1] and Neighbor2Seq [2]. Both methods consider encoding multi-hop neighbors as this work (See Eq. (4)). The key difference and novelty compared to them should be clarified.

2. This method is proved to be at least as equivalent to the expressive power of subgraph GNNs with certain conditions. However, the second condition does not hold for most times. Hence, the expressiveness compared with subgraph GNNs is not convincing to me. In addition, there is no strong empirical results can verify that the proposed method is competitive with subgraph GNNs.

3. Figure 2 is not clear to me. I am wondering if node 1, 2, 3, 4 have the same node features. If yes, then these two regular graphs are isomorphic. If no, then the rooted subgraphs (the middle in Figure 2) can distinguish these two graphs. Hence, I don’t understand why the proposed method is special in this case. Please correct me if I am understanding it wrongly.

[1] Rossi, Emanuele, et al. "Sign: Scalable inception graph neural networks." arXiv preprint arXiv:2004.11198 7 (2020): 15.
[2] Liu, Meng, and Shuiwang Ji. "Neighbor2Seq: Deep Learning on Massive Graphs by Transforming Neighbors to Sequences." Proceedings of the 2022 SIAM International Conference on Data Mining (SDM). Society for Industrial and Applied Mathematics, 2022.


**Summary Of The Paper:**

This paper proposes a simple model to improve the GNN expressiveness by encoding a multi-hop multi-color rooted subtree. It is shown that the proposed method is more expressive than 1-WL GNNs and is efficient in terms of running time and memory usage. Experiments on several benchmarks are performed to support the claim.

**Summary Of The Review:**

Overall, I think this work lacks novelty and experimental support for the main claim. I highly recommend the authors to improve this work by considering the above concerns.

---

### Official Review · Reviewer_Znxn · 2022-10-24

**Confidence:** 4
**Correctness:** 2
**Technical Novelty And Significance:** 2
**Empirical Novelty And Significance:** 2
**Recommendation:** 5

**Clarity, Quality, Novelty And Reproducibility:**

Overall, the paper is clearly written and easy to read. The part around Lemma 1 is a bit hard to understand. I don't see why $\hat{A}_k$ is expensive to compute and how Lemma 1 help reduce the cost to compute it.

Regarding the quality, on the positive side, the overall ideas of coloring the k-hop neighbors and using long RW to expand the receptive field make sense. However, the main issues regarding the quality include:
* missing important related work
* concerns on the correctness of the theorems and complexity analysis (see above)
* missing comparison with subgraph GNNs in evaluation

The architecture design seems novel. I haven't seen existing works performing the same color coding and feature augmentation based on landing probability. The theoretical analysis provides limited novel perspectives. Existing works such as Zhang & Li (2021) and SHADOW-GNN (mentioned above) have already exceeded 1- & 2-WL. It would be interesting to see how long-RW as node features could help further improve the expressivity. Unfortunately, such analysis is missing in the proof (see above).

The authors provide code in an anonymous GitHub repo. However, instructions to reproduce the experimental results have not been provided.

**Strength And Weaknesses:**

## Strengths

+ The paper is clearly written and easy to follow.
+ The overall design is reasonable. Explicit node coloring is somewhat similar to the distance-encoding trick known to the community. The RW-based feature augmentation is a reasonable way to enlarge the receptive field without introducing more message passing steps in the GNN (while at additional cost of preprocessing)
+ Extensive experiments have been performed on both synthetic and real-world datasets.


## Weaknesses

- The description about "subgraph GNN" is a bit confusing. First, a closely related work, SHADOW-GNN [a], is missing. SHADOW-GNN is a recently proposed subgraph GNN for node-level and link-level tasks. It also serves as the backbone architecture for Zhang & Li (2021) and also provably achieves expressivity beyond 1- & 2-WL tests. Since the proposed work is evaluated on both node-level and graph-level tasks, SHADOW-GNN should be properly discussed and compared. Second, the operations / architecture of subgraph GNN have not been clearly defined in the paper. It is hard to tell from Fig 3 what exactly subgraph GNNs do.
- Theoretical analysis seems problematic. First, the example in Fig 2 is wrong. When the nodes have identical initial features (as assumed by the paper), the two 4-node 2-regular graphs are actually isomorphic (the first graph can be transferred to the second by permuting nodes (2,3) to (3,2)). Thus, subgraph GNNs do not fail in this case. Second, the proof of Theorem 2 is at best not rigid. The proof only vaguely says that a long enough RW can return different node features for different subgraph structures. It is not obvious to me why this is always the case for **all** possible subgraph structures. This is an important gap to be filled by the current proof. In addition, even if there do exist a long RW to satisfy the requirement, how long is "long enough"? The length of RW would play an important role in complexity analysis.
- Complexity analysis is problematic. First, as discussed above, $L\ll m$ may not hold since Theorem 2 does not specify the magnitude of $L$. Secondly, even if $L\ll m$, we cannot simply say the complexity of $O(Lm)$ reduces to $O(m)$. For example, if $L=log(m)$, then apparently $O(m\cdot log(m))\neq O(m)$.
- I don't understand the usage of Lemma 1. It seems that you want to address the performance bottleneck of computing $\hat{A}_k$, but why is $\hat{A}_k$ expensive to compute in the first place?
- The experiments are not fully convincing. SOTA subgraph GNNs should be compared with. For example, for the node classification task, the authors should compare with SHADOW-GIN with depth=$L$ and scope=$(L-1)$-hop. For the graph classification task, the authors should compare against Zhang & Li (2021).


## References

[a] Zeng et al. Decoupling the depth and scope of graph neural networks. In NeurIPS 2021.

**Summary Of The Paper:**

This paper presents a GNN design that directly aggregates the multi-hop neighbors by assigning the node features based on the different hop numbers. The explicit node coloring augments the message passing process and thus enables the model to differentiate some non-isomorphic subgraphs that a regular 1-WL will fail to distinguish. In addition, this work proposes a feature augmentation scheme by using the landing probability of long random walks as the additional node features. The RW features helps expand the receptive field of the GNN and further improves the model expressive power. The expressive power is analyzed against 1- and 2-WL tests. Experiments on both synthetic and real-world datasets show performance gains on accuracy and memory consumption compared with the full graph-based GNNs.

**Summary Of The Review:**

In summary, I think the paper is below the bar of acceptance at its current stage. I encourage the authors to improve the theoretical soundness and make the SOTA comparison more complete.

---

### Decision · Program_Chairs · 2023-01-20

**Decision:**

Reject

**Justification For Why Not Higher Score:**

All reviewers raised serious concerns on the paper. This includes the problems with the theoretical analysis, complexity analysis, as well as the experimental settings. The paper also misses important related works, and lacks technical novelty. The conclusion is that the paper is clearly below the bar of ICLR and should not be accepted.


**Justification For Why Not Lower Score:**

N/A

**Metareview: Summary, Strengths And Weaknesses:**

This paper proposes a simple model to improve the GNN expressiveness by encoding a multi-hop multi-color rooted subtree. It is shown that the proposed method is more expressive than 1-WL GNNs and is efficient in terms of running time and memory usage. Experiments on several benchmarks are performed to support the claim.

All reviewers raised serious concerns on the paper. This includes the problems with the theoretical analysis, complexity analysis, as well as the experimental settings. The paper also misses important related works, and lacks technical novelty. The conclusion is that the paper is clearly below the bar of ICLR and should not be accepted.